# Comparative Analysis of Centralized and Federated Learning Techniques for Sensor Diagnosis Applied to Cooperative Localization for Multi-Robot Systems

**DOI:** 10.3390/s23177351

**Published:** 2023-08-23

**Authors:** Zaynab El Mawas, Cindy Cappelle, Mohamad Daher, Maan El Badaoui El Najjar

**Affiliations:** 1CRIStAL, Centre de Recherche en Informatique Signal et Automatique de Lille, University of Lille, CNRS, UMR 9189, F-59000 Lille, France; cindy.cappelle@univ-lille.fr (C.C.); maan.el-badaoui-el-najjar@univ-lille.fr (M.E.B.E.N.); 2Computer Science Department, Beirut Arab University, Beirut 1107, Lebanon; m.daher@bau.edu.lb

**Keywords:** fault tolerance, cooperative localization, machine learning, federated learning, information theory, data fusion

## Abstract

Cooperation in multi-vehicle systems has gained great interest, as it has potential and requires proving safety conditions and integration. To localize themselves, vehicles observe the environment using sensors with various technologies, each prone to faults that can degrade the performance and reliability of the system. In this paper, we propose the coupling of model-based and data-driven techniques in diagnosis to produce a fault-tolerant cooperative localization solution. Consequently, prior knowledge can guide a discriminative model that learns from a labeled dataset of appropriately injected sensor faults to effectively identify and flag erroneous readings. Going further in security, we conduct a comparative study on learning techniques: centralized and federated. In centralized learning, fault indicators generated by model-based techniques from all vehicles are collected to train a single model, while federating the learning allows local models to be trained on each vehicle individually without sharing anything but the models to be aggregated. Logistic regression is used for learning where parameters are established prior to learning and contingent upon the input dimensionality. We evaluate the faults detection performance considering diverse fault scenarios, aiming to test the effectiveness of each and assess their performance in the context of sensor faults detection within a multi-vehicle system.

## 1. Introduction

The increasing usage of robots and autonomous vehicles across various industries has transformed the way automation and mobility are perceived. These advanced machines offer tremendous potential for improving efficiency and productivity. However, their successful operation heavily relies on ensuring their safety and determining with high accuracy their precise position in the environment. This is a very complex task and is crucial for effective navigation, obstacle avoidance, and decision-making.

The necessity of diagnosing and addressing localization faults cannot be overstated. Incorrect localization estimates can lead to collisions, navigation errors, or even dangerous situations in critical applications like autonomous driving. For instance, in May 2016, the National Highway Traffic Safety Administration (NHTSA) investigated a case where a Tesla Model S driver was killed in Florida when their vehicle collided with a tractor-trailer while in autopilot mode. Failure to diagnose and rectify faults has had grave consequences, and with the increase in reliance on these machines, this problem arises further. Thus, a robust and accurate diagnosis of localization issues is essential for ensuring safe and reliable operation.

Various methods were deployed in this field of localization solution diagnosis. They differ given the problem formulation and the main performance required. For the diagnosis of the obtained solution, the creation of fault indicators (residuals) varies given the way the data are perceived and the system requirements. For model-based diagnosis, for instance, fault indicators are computed to express the variation of behavior in the system, and are used as residuals describing the abnormalities. In the deterministic approach, these residuals can be computed using parity space, Luenberger observer, analytical redundancy relationships, and many more methods. For a stochastic non-deterministic approach, however, the data are considered as a probability distribution. In this case, information theory-based indicators are used in order to compare the dissimilarity between these probability distributions. The work of ref. [1] presents a layered architecture for fault detection and isolation in cooperative mobile robots, capitalizing on both analytical redundancy and sensor redundancy. In ref. [2], an Unknown Input Observer (UIO) bank was developed for the detection and isolation of sensor faults, using information theory and the Kullback–Leibler divergence as fault indicators (residuals). Another way to diagnose data is by using the data itself to deduce the behavior. The emergence of artificial intelligence (AI) has revolutionized the way problems are perceived and resolved. AI techniques, such as machine learning and deep learning, have shown great promise in automatically detecting and diagnosing faults and anomalies. By leveraging AI algorithms in the field of diagnosis of localization solutions, it becomes possible to analyze large volumes of sensor data, identify patterns, and detect subtle deviations that may indicate sensor malfunctions or inaccuracies. The integration of AI into the localization process requires a systematic approach, involving data preparation, model training, validation, and deployment. Effective training and implementation of AI models are crucial for accurate fault diagnosis and reliable performance.

On the other hand, multi-robot systems have garnered considerable attention due to their potential for enhanced performance and capabilities. In these systems, multiple robots collaborate and communicate with one another to accomplish shared objectives. The concept of a Cooperative Positioning System (CPS) was first introduced back in 1994 in ref. [3], to deal with the navigation of robots in unknown environments. By exchanging information, validating measurements, and collaborating on localization tasks, the accuracy and robustness of individual robot estimations can be significantly improved. Over time, the field of multi-vehicle systems has advanced in terms of data, architecture, and implementation.

The work in cooperative localization has evolved ever since, studying various aspects. Focusing on communication, ref. [4] presents a distributed message-passing approach for cooperative localization of autonomous mobile vehicles in vehicle-to-vehicle networks. The approach utilizes mm-wave wireless connections to exchange relative distance and angle of arrival information, with the main objective being to estimate all vehicles’ locations through reciprocal exchanges of simple information (messages) in a distributed and autonomous manner. In a map-oriented study, ref. [5] introduces a decentralized cooperative localization method for autonomous vehicles using Local Dynamic Maps (LDMs) based on estimating relative pose using a 2D LiDAR and focusing on ensuring consistency by modeling uncertainty and bounding estimation errors. It also includes a bias estimator to reduce position errors for non-differential GNSS receivers based on visual observations of geo-referenced features in an HD map. Other studies focused on resolving the problem of consanguinity by applying the cooperative solution through the estimator applied. Ref. [6] introduced a fusion algorithm that utilizes a Split Intersection Covariance Filter to handle the correlation between the pose estimations of the beacons for a relative localization relying on low-cost Ultra-Wide-Band (UWB) sensors.

Regarding robot architecture and heterogeneity, ref. [7] focused on decentralized cooperative localization for teams of heterogeneous mobile robots, addressing the challenge of enabling robots with different sensing capabilities to collaborate and estimate their positions accurately. In terms of robot relations and implementation, ref. [8] addresses the challenge of accurately localizing a fleet of vehicles in an environment and updating the map of that environment, by focusing on optimizing the localization process by using a combination of GPS data and visual odometry techniques.

By sharing information, vehicles have the opportunity to consider the opinions of others regarding their position, enabling them to rectify their position and exit a closed-loop state of integration of their own position, which can drift over time. This is particularly valuable in cases of incipient faults, where a vehicle may not be aware that it has encountered such a situation on its own. Conversely, robots can also notify others if they detect a fault in their own system. This is especially beneficial when the vehicles have heterogeneous characteristics, allowing those with less precise sensors to benefit from others to obtain a more accurate position estimation. However, failing to detect a fault within a vehicle can result in the propagation of this error throughout the entire network of vehicles. This highlights the importance of well-organized data resources and classifiers.

As systems become increasingly complex, the need for advanced diagnostic techniques arises. One approach that has gained considerable attention in the field of diagnosis is the hybridization of model-based techniques and data-driven approaches. Models capture the underlying physical laws, system dynamics, and interdependencies between different components of the system. This structured representation enables model-based techniques to provide valuable context and domain knowledge to the data-driven techniques. By incorporating the models into the diagnostic system, data-driven algorithms can be guided by the underlying system behavior and constraints. Moreover, by comparing the expected system behavior predicted by the models with the actual observed behavior, model-based techniques can identify deviations and anomalies. These deviations can guide the data-driven techniques by narrowing down the search space and focusing their analysis on specific areas of interest.

A very early implementation of such techniques for sensor fault detection was in ref. [9], where multiple parallel Kalman filter estimators were deployed, each incorporating models of the system behavior corresponding to different fault types, coming from either sensors or actuators. From those estimators, residuals were computed based on the Mahalanobis distance, and used as input to a back-propagation Neural Network, processing this set of residuals as a pattern. Also, in ref. [10], a state fault detection in autonomous vehicles is presented using One-Class Support Vector Machine (SVM). It trains a boundary curve that effectively separates the safe and unsafe domains of the vehicle’s behavior, which current position is predicted using a Kalman filter observer designed using the linear kinematic vehicle bicycle model. These techniques were also used in the field of multi-robot systems. In ref. [11], an algorithm that combines model-based and data-driven methods to identify an optimal set of residual generators was introduced, with a generalized diagnosis system design utilizing multi-variate residual information to enhance fault detection. Moreover, ref. [12] proposes a modified fusion model for sensor fault diagnosis and performance degradation estimation in aero-engines, combining on-board models and a data-driven model for residual correction. Its framework includes a bidirectional information transmission algorithm to facilitate function coordination, with a Kalman filter as the optimal algorithm, with a standardized sensor parameter selection process. Another paper combing model-based and data-driven approaches is ref. [13], where an adaptive Kalman filter was combined with the generalized likelihood ratio method and followed by a data-driven gap metric strategy to address component faults, especially incipient ones with disturbances or noises.

However, handling data even in a homogeneous multi-robot system with different trajectories is not a straightforward task. Organizing the data and studying how various resources and their history affect the behavior of the system is crucial. In ref. [14], a heuristic analytical model for residual with threshold determination using the ROC curve was implemented. This method was able to improve the localization solution, but was limited by the need for prior knowledge of intrinsic faults of sensor measurements. To overcome this limitation, the paradigm was changed from model-based diagnosis to data-driven diagnosis based on AI tools. A first method was applied in ref. [15] with a centralized unit that gathers information from different robots with different trajectories navigating in the same environment, to train a Multi Layer Perceptron (MLP) with residuals based on Jensen–Shannon divergence and history indicators based on the prior probability of the no-fault hypothesis $P_{0}$. The main objective of this work is to apply a model-based/ data-driven sensor fault detection and exclusion in a decentralized cooperative localization system under centralized and federated learning techniques. The model used for this study is the Logistic Regression Classifier, given that it is a classifier with parameters proportional to the number of inputs, and is able to identify which ones do not contribute to the classification task. We compare its performance in centralized and federated learning to a system composed of three robots. The study is limited to three robots in the validation of the cooperation hypothesis of having ≥2 robots, and adding one robot above this limit to validate the hypothesis of expansion. The analysis performed in this work seeks to assess the influence of subsystem behavior on the generated models and the final aggregated model, as well as to evaluate the performance of these models on the individual subsystems.

The paper is organized as follows: this introduction, then problem formulation, and assumptions. The third section presents the proposed approach, starting with the generation of data, its exploitation, and training, presenting the chosen model to apply the method. Next, the results are shown and discussed for three types of implementation: centralized approach, models optimized on the robot itself, and federated learning. Finally, a conclusion and perspectives are proposed.

## 2. Problem Formulation and Assumptions

The problem at hand revolves around developing an efficient and reliable localization solution that heavily relies on sensor measurements, and that is able to check if they are valid or not. In a multi-sensor localization system, sensors exhibit a natural redundancy since they consider the same information but utilize different technologies to observe the surrounding environment. This redundancy serves as a beneficial aspect as it enhances the overall robustness and accuracy of the localization system. In addition, each sensor has its own inherent limitations and uncertainties associated with its measurements. These uncertainties can arise due to various factors such as sensor quality, environmental conditions, or inherent technological limitations. Therefore, the selected position estimation method should be capable of effectively fusing data from multiple sensors while taking into account their stochastic nature. To answer the nature of data, divergences which are information theory dissimilarity measurements can be used.

Outside the bounds of their uncertainties, sensors pass to abnormal situations. These faults can stem from various sources, including calibration issues, physical damage, or interference from external factors. When errors occur, they can propagate through the system and can have a significant impact on the overall accuracy and reliability of the localization solution. And as mentioned before, the precision of the localization solution is necessary given that it is the input to the autonomous navigation and path planner algorithms. Therefore, it becomes crucial to develop a method capable of discriminating between system functioning states and pinpointing the sensors that are faulty.

The localization system can be effectively represented by a model-based method that describes its expected behavior. However, when a fault occurs, the system’s performance deviates from the expected behavior, making it challenging to accurately predict its behavior or classify its current functioning hypothesis over time. In such cases, the deployment of adaptive thresholding becomes necessary. By utilizing the equations derived from the model, fault indicators can be obtained by comparing the system’s actual performance with its expected performance based on previous states and other redundant data entries.

To address this challenge, machine learning techniques offer a promising solution to model the system’s behavior in an adaptive way. To classify between functioning hypotheses, an effective model needs to be deployed—one that is capable of discerning between normal and faulty scenarios. This falls under the capabilities of discriminative models that focus on explicitly modeling the boundaries between different classes and aim to classify new instances based on these boundaries. By adopting such an approach, the model can provide valuable insights into the presence and nature of faults within the system.

To test and validate the effectiveness of thus handling the problem at hand, we set up an experimental platform with the following assumptions and limitations:Proprioceptive and exteroceptive sensors are equipped on *N* robots, where *N* = 3 in this work.Each robot is able to observe at least one other robot and each robot is at least observed by another one.At least two sensors must be functioning to ensure the obtainment of a localization solution. Otherwise, no localization solution can be computed.Vehicles are homogeneous in types and technologies.Architecture for data fusion is fully decentralized.Communication between the robots is maintained at all times (the WiFi is used in this work).Sensor faults can be assimilated to erroneous measurement, confusion between robot and another object, bias, outlier measure, and absence of information.

The aimed contribution of this work is to create a method that benefits from both model-based and data-driven diagnosis methods to compensate for the limitation of each in order to increase the precision of the system in various scenarios. It also studies the possibility of decentralizing the system in learning as well as data fusion, and compares the impact of this on the system. In fact, the limitation of model-based techniques is that they heavily rely on accurate and comprehensive mathematical models of the system under consideration. Deviations or uncertainties in these models can lead to inaccurate diagnoses, especially in an erroneous situation that can accumulate. Furthermore, model-based approaches may struggle when dealing with novel or unforeseen situations that fall outside the scope of the established model, potentially leading to missed diagnoses. On the other hand, complex data-driven models, such as deep neural networks, often lack interpretability, making it challenging to understand the reasoning behind a diagnosis. Combining these two approaches can harness their strengths, for by integrating model-based techniques with data-driven methods, the model’s assumptions and the system’s behaviors can be cross-validated.

## 3. Proposed Approach

Our method has two parts: data generation (Section 3.1) and data exploitation (Section 3.2). The data generation step signifies the application of the cooperative localization solution and provides the indicators that are considered as the data input for the learning algorithms. The data exploitation step presents the possible learning techniques, then the selection of the classification model, its elements, and training.

### 3.1. Data Generation

The data generation step’s aim is to compute model-based fault-sensitive indicators to guide the data-driven technique. It has the following four steps:**Data acquisition:** The data are acquired from three different types of sensors:Prediction: Wheel encoder applying *odometry* as evolution model;Correction: *Gyroscope*, *Marvelmind*, and relative observations that are computed from the predicted state of the other nearby robots and the *LiDAR* measurement.Five trajectories having different shapes have been recorded using three Turtlebot 3 burger robots under Robot Operating System (ROS). They are presented in Figure 1. These trajectories were generated to encompass scenarios in which multiple robots can coexist within an environment and mutually observe one another. The selection of circular or transversal shapes for these trajectories represents the fundamental geometric forms underlying their structure. The data from the embedded sensors goes through a pre-processing stage, during which any faults detected in the sensor data with regard to the ground truth are corrected.**Faulty scenarios generation:** For each of those trajectories, we generate 10 different sensor fault scenarios, taking in consideration the type of sensor and the fault that can occur on it: *drift* for encoders and gyroscope, *bias* for *Marvelmind* and LiDAR, creating a total of *1031 m* and about *an hour* of generated data for each robot.**Position and orientation estimation:** Applying the prediction step and correction step of the Extended Informational Filter (EIF), a variation of Kalman filter with identical prediction step, but where the correction is obtained by the summation of the informational contributions of the sensors in the information space. To do so, the covariance matrix of the prediction step $P_{k|k-1}$ and the state vector $X_{k|k-1}$ should be projected to the information space by applying:
(1)$$\left\{\begin{array}{c} Y_{k|k-1}=P_{k|k-1}^{-1}\\ y_{k|k-1}=Y_{k|k-1}\times X_{k|k-1} \end{array}\right.$$$Y_{k|k-1}$ and $y_{k|k-1}$ represent the information matrix and vector, respectively. An observation is defined by $\left\{Z_{obs},H_{obs},R_{obs}\right\}$, where $Z_{obs}$ is the measurement vector, $H_{obs}$ is the observation matrix and $R_{obs}$ is the measurement noise covariance matrix. The maximal number of observations is L=S+(N−1) observations, where *S* is the number of exteroceptive sensors and (N−1) is considered as the maximum number of observations that can be obtained from the LiDAR toward the robots in range. The correction step is applied as the following [16]:
(2)$$\left\{\begin{array}{c} P_{k|k}=(Y_{k|k-1}+\sum_{i=1}^{L}I_{obs^{i}}){}^{-1}=(Y_{k|k-1}+\sum_{i=1}^{L}\left(H_{obs^{i}}^{T}\times R_{obs^{i}}^{-1}\times H_{obs^{i}}\right){)}^{-1}\\ X_{k|k}=P_{k|k}\times (y_{k|k-1}+\sum_{i=1}^{L}i_{obs^{i}})=P_{k|k}\times (y_{k|k-1}+\sum_{i=1}^{L}\left(H_{obs^{i}}^{T}\times R_{obs^{i}}^{-1}\times Z_{obs^{i}}\right)) \end{array}\right.$$
where $I_{obs^{i}}$ and $i_{obs^{i}}$ are the informational contributions of the used observation.**Information theory based faults indicators generation:** The generated residuals are based on the Jensen–Shannon divergence between two probability distributions, a symmetrization of the Kullback–Leibler divergence. For two distributions *p* and *g*, the Jensen–Shannon divergence is:
(3)$$D_{JS}\left(p\left(x\right)\right|\left|g\left(x\right)\right)=h\left(M\right)-\frac{1}{2}\left(h(p(x))-h(g(x))\right)$$
where $M=\frac{1}{2}\left(p\left(x\right)+g\left(x\right)\right)$ is the mean distribution of *p* and *g*, and $h\left(p\right)=\sum_{j}p_{j}log\left(\frac{1}{p_{j}}\right)$ is the entropy [17].For each robot having *L* observations, $\left(1+\sum_{j=1}^{L-2}C_{L}^{L-j}\right)$ indicators are generated, one generalized, implementing all the observations in the correction step. The $C_{L}^{L-j}$ implies the isolation residuals, under the Generalized Observer Scheme (GOS), where each time, a combination of residuals where one observation is removed from the correction step, then two, until we have at least two sensors that are operating. For our case, where we have L=2+(3−1)=4 observations, (1+6+4) residuals = 11 type of residuals are generated for each robot. One for detection: the Generalized Jensen–Shannon divergence $gD_{JS}$ and 10 for isolation $JS^{obs_{GOS}^{a}}$ following the Jensen–Shannon divergence as well under Generalized observer Scheme (GOS), with $obs_{GOS}^{a}\in \{m^{a},g^{a},L^{a\leftarrow b},L^{a\leftarrow c},\left[m^{a},g^{a}\right],\left[m^{a},L^{a\leftarrow b}\right],\left[m^{a},L^{a\leftarrow c}\right],\left[g^{a},L^{a\leftarrow b}\right],\left[g^{a},L^{a\leftarrow c}\right],$$\left[L^{a\leftarrow b},L^{a\leftarrow c}]\right\}$.

Algorithm 1 presents the evolution of the approach in terms of data generation, and the precise use of the classifiers for the detection and isolation of sensor faults.
**Algorithm 1** Fault tolerant Cooperative Localization algorithm**Require:** Initialization of state with $X_{o}^{a}$ and its covariance $P_{o}^{a}$ obtained from *Optitrack* for each robot.**Require:** Set initial values of $P_{0}$ and $P_{0}^{obs^{a}}$.    **while**
$k\neq Nb_{iterations}$
**do**        * ***Apply prediction step***:        Read data from encoders: $u^{a}\leftarrow {\left[\Delta^{a},\omega^{a}\right]}^{T}$        Compute $\left\{X_{k|k-1}^{a},P_{k|k-1}^{a}\right\}$ using the previous state and the evolution model.        Compute the information vector and matrix $\left\{y_{k|k-1}^{a},Y_{k|k-1}^{a}\right\}$ by using (Equation 1).        * ***Apply correction step***:        Compute the informational contribution {$i_{k}^{obs^{k}}$, $I_{k}^{obs^{k}}$} for $obs^{a}\in \left\{m^{a},g^{a}\right\}$, by using (Equation 2).        **if** $Robot_{b}$ or $Robot_{c}$ in sight **then**            Get their position and covariance $\left\{X_{k|k-1}^{b,c},P_{k|k-1}^{b,c}\right\}$            Get the relative observation {$Z_{k}^{L^{a\to b,c}},R_{k}^{L^{a\to b,c}}$} toward them.            Compute the correction {$X_{k}^{obs^{k}}$, $P_{k}^{obs^{k}}$} for $obs^{a}\in \left\{L^{a\to b},L^{a\to c}\right\}$.        **end if**        Compute {$X_{k|k}^{a}$, $P_{k|k}^{a}$} using the Extended Information filter in (Equation 2).        * ***Apply Diagnosis step***:        Compute $gD_{JS}^{a}$ using (Equation 3).        **if** $\left\{gD_{JS}^{a},P_{0}\right\}$ tested on the detection model implies the presence of a fault **then**            Compute $JS^{obs_{GOS}^{a}}$ using (Equation 3)            Get the faulty sensors using the isolation models            Exclude the faulty sensors            Update the position and covariance {$X_{k|k}^{a}$, $P_{k|k}^{a}$}        **end if**        Update $P_{0}$        Update $P_{0}^{obs^{a}}$        *k = k + 1*    **end while**

It first sets the initial values and dimensions of the state vector and the variance-covariance matrix. In this work, the state vector is composed of the coordinates ${\left[x\;y\;\theta \right]}^{T}$, the position on the x-axis and y-axis, and the orientation of the z-axis θ. This means that the state vector of $robot_{a}$ for example $X_{o}^{a}$in$R^{3\times 1}$, and its variance-covariance matrix $P_{o}^{a}$in$R^{3\times 3}$. The initial values of these coordinates for each robot are set by their value in the used ground truth the *Optitrack*. Given that the performance with respect to errors is recorded for the general behavior of the system and for each sensor, an initial value is set for the prior no-fault probability hypothesis $P_{0}$.

After the initialization phase, the main loop of the algorithm is applied. First, the prediction step of the information filter is applied. It uses the odometric model based on encoder data to predict the position of the robot based on the evolution it has made, using the input vector $u^{a}=\left[\Delta^{a}\;\omega^{a}\right]$, the elementary displacement and the rotation performed during the iteration when it is computed.

After the prediction, the projection into the information space is applied, using (Equation 1), in preparation for the correction step. If the correction is using an embedded sensor observing the robot itself, the informational contribution vector and matrix are computed taking into consideration the sensor’s parameters, such as the uncertainty and the observation matrix. Otherwise, if the correction comes from the LiDAR, given that other robots in the system are in sight, the portable landmarks are applied. These robots in sight broadcast their position and variance-covariance matrix in the shared network between the robots. The robot observing other robots retrieves this position, and along with LiDAR observation indicating the relative position of this robot, computes the informational contribution of this observation. When all the available informational contributions are computed, the correction model is applied as indicated in (Equation 2), summing these values with the information vector and matrix appropriately and then projecting back to the Kalman space.

Once the correction is obtained, the robot proceeds to diagnosing the position. This is performed using information theory, and by computing the dissimilarity by the Jensen–Shannon divergence (Equation 3) between the prediction and correction at first to know if there is a fault in the system overall, which constitutes the detection step. The determination of the functioning hypothesis is performed using the pre-trained detection model. If this model indicates the presence of a fault, the isolation step is applied. As mentioned before, the Generalized observer Scheme (GOS) is applied to compute fault indicators, which, along with the value of the prior no-fault hypothesis of sensors are fed to each isolation model, each dedicated for a sensor, and indicating the functioning hypothesis of each as output. Using the output of these models, the position and covariance are updated by excluding the indicated faulty sensors from the fusion step. Finally, the values of the prior no-fault hypothesis are updated given the findings of the current iteration.

### 3.2. Data Exploitation

In order to be able to detect and exclude faults, two types of models are implemented: one unique for detection and multiple ones for isolation. These models are trained in two different ways: centralized where all the data are sent to a central unit, and federated where the data are trained locally and remains on the machine, and only the models are communicated to the central unit in order to be averaged.

#### 3.2.1. Data Organization

In a system where the data are produced by various members and in order to train a model, there are many ways to organize the data. In this work, we focus on two of them: centralized and federated learning.

In the centralized data organization method, all the data from the network members is collected and consolidated into a central location. The learning model is then trained using this centralized dataset, then provided to each member. This method is beneficial due to its ease of implementation and utilization of various training algorithms. Moreover, it allows for comprehensive analysis and processing of the entire dataset, and can potentially achieve better performance if the centralized dataset is diverse and representative. However, it requires transferring data from individual members to a central location, which may raise privacy and security concerns, and it bounds the functioning of the system with the responsivity and availability of the central unit. It also assumes that the centralized dataset represents the entire population and may not account for local variations.

Changing the architecture to a decentralized one, the learning becomes federated. This approach allows training a model directly on distributed data without the need to transfer the raw data to a central location. In federated learning, the model is deployed on each network member, and training is performed locally on their respective data. The model’s parameters are then periodically aggregated or averaged across the network to create a global model that benefits from the collective knowledge while preserving data privacy.

This approach preserves data privacy and security by training the model locally on each member’s data without transferring raw data. It also allows learning from a large number of distributed devices or nodes, enabling broader participation, and allows reducing communication overhead as only model updates or gradients are shared, rather than raw data. However, it requires synchronization and aggregation of model updates, which can introduce additional complexity. In addition, it may face challenges with heterogeneous data distribution across network members, leading to variations in model performance, especially when systems have different histories of functioning, imposing that their behavior is somewhat different from the others, therefore not adapted to the rest of the system. The quality of the global model in this method also depends on the participation and quality of data from all network members.

#### 3.2.2. Model Optimization and Training

The problem of classification requires discriminative learning, under which, there are many models that can be used in Machine Learning. In order to be able to compare the performance between centralized and decentralized techniques, under the federated learning aggregation methods, we select the logistic regression model. Unlike other models, and in order to apply federated learning, a limited number of parameters known to all subsystems is used, given that the input to the models is the same.

Logistic regression or logit models are a special case from the Generalized Linear Model (GLM), for binary and multi-class classification tasks. It provides a probabilistic framework to estimate the likelihood of class membership based on input features [18]. Although this tool was first introduced in ref. [19] in the mid-20th century, its implementation was delayed due to the limitations of computational resources, the availability of alternative algorithms, and a gradual recognition of its effectiveness.

Logistic regression provides a mechanism for applying the techniques of linear regression to classification problems, but by changing the space where these values are computed. It utilizes a linear regression model of the form
(4)$$z=\beta_{0}+\beta_{1}x_{1}+\ldots +\beta_{n}x_{n}$$
where $x_{1}$ to $x_{n}$ represent the values of the n attributes or input variables and $\beta_{0}$ to $\beta_{n}$ represent weights or intercept. This model is mapped onto the interval [0, 1] using the logit function defined as follows:(5)$$P(c_{0}\left|x_{1},x_{2},\ldots ,x_{n}\right)=\frac{1}{1+e^{-z}}$$

β affects the logit (logarithm of the odds). Logistic regression parameters refer to odds and odds ratios. For continuous input variables, the log-likelihood ratio is to be optimized to have a maximum value in order to obtain the best-fitted curve for classification.

Logistic regression is also used to estimate the relationship between a dependent variable and one or more independent variables, but it is used to make a prediction about a categorical variable versus a continuous one.

To best fit a logistic regression model, the fitting parameters are the following:Penalty: determines the type of regularization applied to the logistic regression model. Regularization helps prevent over-fitting by adding a penalty term to the loss function. Two methods exist: L1 and L2.L1 Regularization (Least Absolute Shrinkage and Selection Operator Regression − Lasso Regularization): it adds a regularization term to the cost function, but it uses the $l_{1}$ norm of the weight vector.L2 Regularization (Ridge Regularization): a regularization term equal to $\alpha \sum_{i=1}^{n}\theta_{i}^{2}$ is added to the cost function. This forces the learning algorithm to not only fit the data but also keep the model weights as small as possible [20].C: is the inverse of the regularization strength. It controls the extent of regularization applied to the logistic regression model. A smaller value of C increases the regularization strength, resulting in a more constrained model with smaller coefficients.Solver: specifies the algorithm used for optimization during model training. Various factors are considered by different solvers, such as the dataset size, convergence speed, support for regularization, linearity of data, memory efficiency, constraints, parallelization capability, Hessian approximation, and whether stochastic or batch updates are usedWeight class: The data are imbalanced and if the model is trained using the data as it is, it will be biased toward the abundant class. To prevent this, cost-sensitive learning is deployed. It takes into account the cost of miss classifying different classes in a multi-class classification problem. In this case, a weight must be accorded to each of the labels used.

Given that we want to keep all parameters, we use the L2 Regularization. We fix C to 0.01 and we use Limited-memory Broyden-Fletcher-Goldfarb-Shanno (lbfgs), the default solver used for logistic regression.

The aggregation of data can be in many ways, the simplest one is the Federeated Averaging (FedAvg) method in ref. [21]. In this method, having *K* clients, each with a subset $D_{k}$ of length $n_{k}$, the aggregation over the hyper-parameters ω of the models is performed by applying the following equation:(6)$$\omega_{t+1}=\sum_{k}^{K}\frac{n_{k}}{\sum_{k}^{K}n_{k}}\times \omega_{t+1}^{k}$$
In our case, for the detection model, two variables are used: $P_{0}$ the prior probability of the no-fault hypothesis and $gD_{JS}$ the detection residual following Jensen–Shannon divergence, meaning that the equation has three parameters, the intercept $\beta_{0}$ and slopes $\beta_{1}$ and $\beta_{2}$, with one output, the existence of a fault.

As for the isolation, we have slopes $\beta_{i}$ for 10 variables that are the isolation residuals $JS^{obs_{GOS}^{a}}$, the $P_{0}$ of each sensor, along with the intercept, resulting in 10 + 4 + 1 = 15 parameters to be optimized.

In a system with four sensors, where faults can occur simultaneously, there are six distinct output classes. These classes include no fault (0), individual faults for each sensor (4 classes), and a separate class for simultaneous faults. Due to its unique characteristics, this model cannot be treated as a standard "one against all" classifier, and thus it cannot be generalized into a single model. Additionally, not all inputs are applicable to all models, as the residuals are generated using the Generalized Observer Scheme. These inputs are fed in the same order as the $obs_{GOS}^{a}$, appended to it the $P_{0}$ of *encoder*, *marvelmind*, *gyroscope* and *LiDAR*, respectively.

## 4. Results

In order to validate the method, 10 scenarios were used, with 1 original and 10 generated ones. Among these scenarios, 8 scenarios were used for training, 0 to 7, and 3 for testing, 8 to 10. The proposed method is presented in Figure 2 showing the different steps of the approach. Two learning techniques have been deployed: centralized learning presented in Figure 3 and Federated learning presented in Figure 4. The three robots used have the same number of samples, which implies that the Fedearated Averaging (FedAvg) algorithm used for parameter aggregation is reduced to a factor of 0.33.

The generation of data and trajectory evolution figures took 290.7094 s, for 5 trajectories of 3 robots with 11 scenarios each. The database occupies 4.6 MB, and the machine used to perform the training is a 16 GB RAM core-i5 PC under Ubuntu 20.04 LTS operating system. The chosen severity for this study is 30%, and the duration of fault is between [1%,3%] of the total time of the trajectory, making the faults brief and low. The scenarios were generated in a way for the sensor faults to occupy different instances each time, and in different orders of occurrence. We consider the measures for comparison up to the fourth decimal place $\left(10^{-4}\right)$, as this is where most of the significant changes occur and provides a meaningful basis for comparing the resultant numbers. The chosen comparison metrics for this study are *accuracy* and *f1-score*, to measure how well the model correctly predicts the class labels of the samples in the dataset, and combine in a single metric both precision and recall to provide a balanced measure of a model’s performance, especially when dealing with imbalanced datasets or scenarios where false positives and false negatives have different implications.

The fitting of the detection model takes 0.0762 s for the centralized learning, while for each robot alone, it takes around 0.01 s. These computations are performed in parallel, meaning that the fitting in federated learning is faster than in centralized learning. The accuracy of the fitted centralized learning model is 0.9463, and the f1-score is 0.7035. For the locally fitted models, $robot_{1}$ model has an accuracy of 0.9489 and f1-score of 0.7189, $robot_{2}$’s accuracy is 0.9462 and f1-score is 0.6144, and $robot_{3}$ has an accuracy of 0.9482 and f1-score of 0.6990. One can notice that the accuracy of each model alone is equal to or higher than those of the centralized learning model. For the f1-score, it is higher in the case of centralized learning than in most of the robots. This is due to having access to a larger and more diverse dataset since data from all clients are combined. This can result in a more representative and comprehensive training dataset, enabling the model to learn patterns and generalize better across various cases. For the federated learning detection model, the accuracy over the trained data is 0.9830 which is higher than any other fitted model. This means that the federated model is more accurate in classifying instances in the dataset. However, despite having better accuracy, the F1 score of the federated model is significantly lower than the F1 score of the centralized model. The F1 score considers both precision and recall, and a lower F1 score indicates that the federated model is not performing well in balancing false positives and false negatives.

Concerning the logistic regression parameters, $\beta_{1}=0.1247$ with respect to the detection residual $gD_{JS}$ and $\beta_{2}=-0.2036$ for the prior probability of the no-fault hypothesis $P_{0}$, meaning that it is inversely proportional to its evolution. The federated learning’s parameters however are $\beta_{1}=0.1355$ and $\beta_{2}=-0.0804$, with a similar value of $\beta_{1}$ to that of the centralized learning, but a lower value for $\beta_{2}$, meaning that the relationship between $P_{0}$ and the occurrence of a fault is weaker in the federated learning model. Regarding memory usage, both models occupy minimal space, with the centralized model requiring 849 bytes and the federated model only using 508 bytes.

The isolation step has a model for each sensor, having all the same input, with 0.08sec for fitting in centralized learning and an average of 0.03 s on each robot. They occupy together a total of 2.2 kB and 1.4 kB for centralized and federated learning, respectively.

The fitted centralized learning model for the *encoder* has an accuracy of 0.9687 and an f1-score of 0.5275. For $robot_{1}$, its accuracy is 0.9811 and f1-score is 0.566, $robot_{2}$ accuracy is 0.9834 and f1-score is 0.5753, and $robot_{3}$ accuracy is 0.9782 and f1-score is 0.5143.

For the *marvelmind* model, the centralized model has an accuracy of 0.9611 and an f1-score of 0.6680. For $robot_{1}$, the accuracy is 0.9606 and f1-score is 0.6262, $robot_{2}$ has an accuracy of 0.9612 and f1-score of 0.6219, and $robot_{3}$ has an accuracy of 0.9619 and f1-score of 0.6418. The locally fitted models have very similar performance to that of the centralized one, with less time consumed.

Concerning the *gyrocospe* model, the fitted centralized model has an accuracy of 0.9770 and an f1-score of 0.7554. For $robot_{1}$ has an accuracy of 0.9715 and f1-score of 0.7154, $robot_{2}$ accuracy is 0.9793 and f1-score is 0.7837, and for $robot_{3}$ accuracy is 0.9836 and f1-score is 0.7896. The performance of the local models is very close to those of the centralized learning model.

Finally, for the *LiDAR* model, the fitted centralized model has an accuracy of 0.9834 and an f1-score of 0.8010. For $robot_{1}$, the accuracy is 0.9805 and f1-score is 0.7916, $robot_{2}$ accuracy is 0.9847 and f1-score is 0.8050, and $robot_{3}$ accuracy is 0.9825 and f1-score is 0.7835. The performance of the local models is also very close to those of the centralized learning model.

The fitting of the parameters used L2-regularization, keeping all the inputs in order to apply the federated learning algorithm. Their values reflect the interest of the input given the sensor studied: in the case of a fault on the *encoder* sensor, all 10 concerning the isolation residuals have balanced close values, which is expected due to the concept of the Generalized observer scheme upon which those residuals were designed. The odometric model which is used as a prediction model appears on each correction, which makes it logical that all the indicators should react if an error occurs on this measure.

In the case of fault over the other sensors used for correction, we notice that the parameters concerning the sensor, which is the prior no-fault probability hypothesis and the isolation residuals that incorporates the faulty measurement gain a higher value than the others, in the centralized as well as each robot’s model. This behavior is expected, given that the residuals were built to reflect and permit the faults to be separated and distinguished, under the generalized observer scheme. For instance, in the case of a fault on the *marvelmind* sensor, the parameter $\beta_{1}$ reflecting the residual that implements all the corrections but that of the *marvelmind*, it has the lowest value among all the other first layer of residuals excluding one sensor at a time. On the other hand, the parameter $\beta_{12}$ that concerns the prior no-fault probability hypothesis of the sensor itself has the highest value among all the other parameters of the same type for the other sensors. This proves that the faults are independent from one sensor to the other. The same analogy is found for the other sensors as well.

For one of the tested scenarios, a table comparing the behavior of the types of classifiers for detection and isolation under the two proposed learning techniques is presented in Table 1 as well as the locally fitted models on each robot. These performances are presented for the 5 trajectories in use, and for the testing scenario=8. The indication $traj_{1}D-Centr$ designates the first trajectory, *D* for detection, and Centr is to indicate the centralized learning model. On the other hand, $traj_{1}D-Fed$ designates that of the federated learning model. The indication $traj_{1}D-Robot$ means the models fitted locally and tested locally on the testing data of the robot itself. For the isolation, a mean of the performances across sensor models is presented. This table presents as discussed regarding the fitting performance that the federated and the locally fitted models have performances equal to or better than that of the centralized learning model. We notice that the performance is similar and sometimes better in federated learning than in the local model over the robot or the centralized one.

Concerning the diagnosis step, the detection residuals for each robot are shown in Figure 5. We chose to show scenario=8 for $traj_{5}$ which has close-by faults and where odometry faults were injected. For $robot_{1}$, the first injection of marvelmind and gyroscope at unix time 1.622546125 are very close, and one can see that the generalized residual reacts to all the faults, but differently. For the marvelmind fault, the fault does a first jump when the fault is applied, returns to its position then does a second jump when its fault is unlatched. After that, a fault over the gyroscope is applied, which has less impact over the detection residuals in this case of trajectory. The detection of a fault triggers the isolation step. The isolation residuals are designed in a way to be able to isolate the detected faults. The first layer of the isolation residuals (with one sensor removed each time) is shown in Figure 6, where their naming with the name of the sensor indicates that it incorporates all the observations except this sensor, following the Generalized observer Scheme (GOS). For the Marvel residual for Robot 1 (Figure 6 first row first column), it is sensitive to the encoder, gyroscopes, and LiDAR faults. The gyroscope residual (second column) is sensitive to the encoder, marvelmind, and LiDAR faults. Finally, the LiDAR residuals (third and fourth columns) are sensitive to the encoder, marvelmind, and gyroscope faults. The same analogy can be made for the other two robots.

We can see the improvement of the trajectory in Figure 7, and the impact of the exclusion of faults on the improvement of the trajectory. Comparing the excluded sensors one by one between centralized and federated learning, we obtained the same faulty sensors detected and excluded in exactly the same instances. This is expected due to the very similar performance between the centralized and federated learning as discussed in the results of Table 1.

## 5. Conclusions and Perspective

In this paper, our study focused on evaluating the impact of federated learning on the performance of machine learning classifiers for a fault-tolerant cooperative localization solution involving three vehicles. It was applied under decentralized multi-sensor data fusion architecture, where each robot computes locally its solution, and does not share raw information with other robots. The decentralization was expanded further in the choice of data organization and learning which is performed locally on each robot, with sharing only their fitted models with the central unit, therefore maintaining the privacy. This increases the robustness and availability of the system, given that the robots are able even to create their own classifiers even if the central unit is out of reach, and makes their solution more exhaustive when aggregating their model with other robots, giving them a wider view of the cases that may occur leading to faulty functioning. Moreover, this method is hybrid, as it benefits from the model-based diagnosis’s ability to provide valuable context and domain knowledge in order to create fault-sensitive indicators, that are used with data-driven techniques. The results demonstrated that the performance of the federated model was comparable, and in some cases, even superior to that of centralized learning. Moreover, the method exhibited a high accuracy in detecting and isolating injected faults. The method was able to detect and exclude the faulty sensors, providing a better and more precise localization solution, and therefore reducing the impact of fault propagation through the system.

Building upon the findings of this research, there are several ways for further exploration and expansion. One potential direction is to incorporate additional sources of information for localization, such as cartography in 2D/3D, inertial navigation systems (INS), and vision data. Integrating these diverse sources increases the complexity of the system. Studying the limitations of this method, and the types of sensors that it could be implemented with is another future direction.

Additionally, the work can be extended by using a dedicated observer scheme to see if this attempt to reduce classifier inputs could maintain the same performance. The generalized observer scheme (GOS) implies the existence of many inputs per observation, with the count of all sensor classifiers using most residues except one for isolation. Such a scheme is used in order to maintain homogeneity between the various residual levels, by keeping the same quantity of information. In contrast, the dedicated observer scheme (DOS) has one observer per sensor, allowing tailored limits based on individual sensor behavior and information at each iteration.

## Figures and Tables

**Figure 1 sensors-23-07351-f001:**
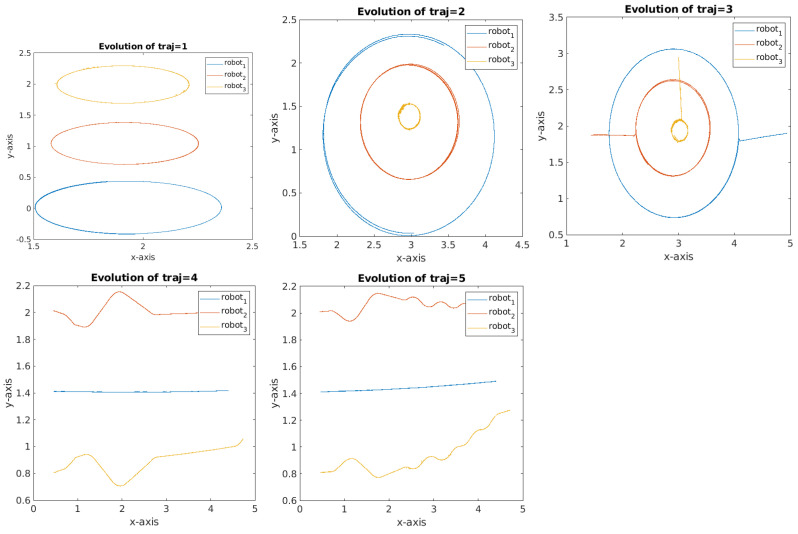
The generated trajectories for learning.

**Figure 2 sensors-23-07351-f002:**
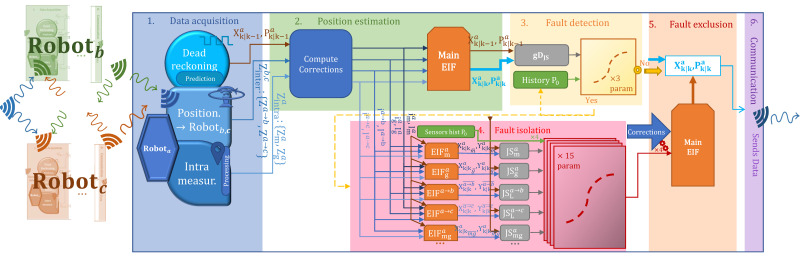
Proposed fault-tolerant cooperative localization technique.

**Figure 3 sensors-23-07351-f003:**
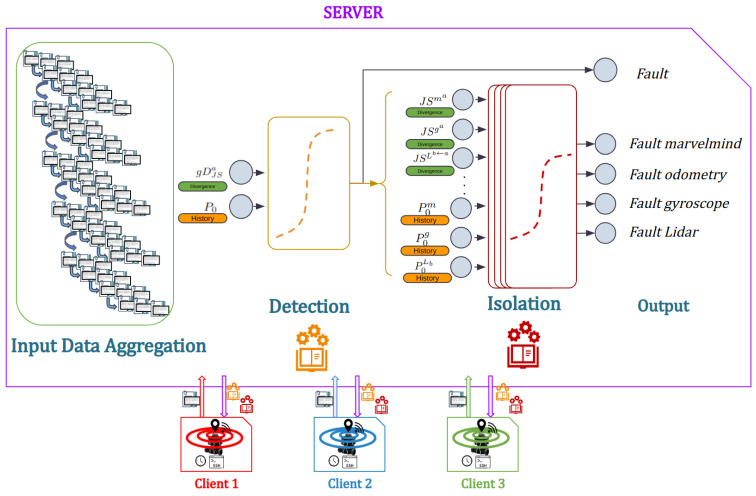
Centralized Learning scheme.

**Figure 4 sensors-23-07351-f004:**
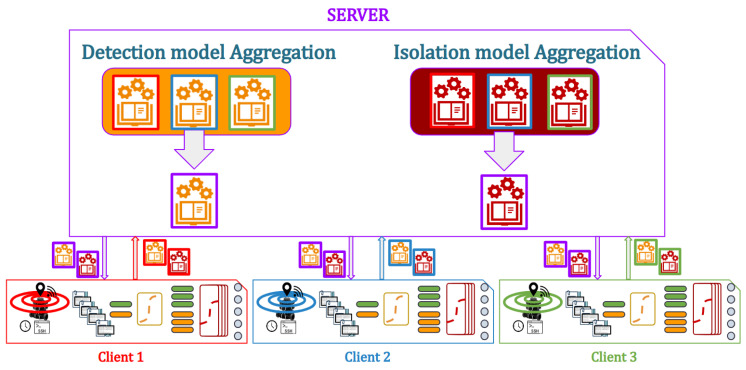
Federated Learning scheme.

**Figure 5 sensors-23-07351-f005:**
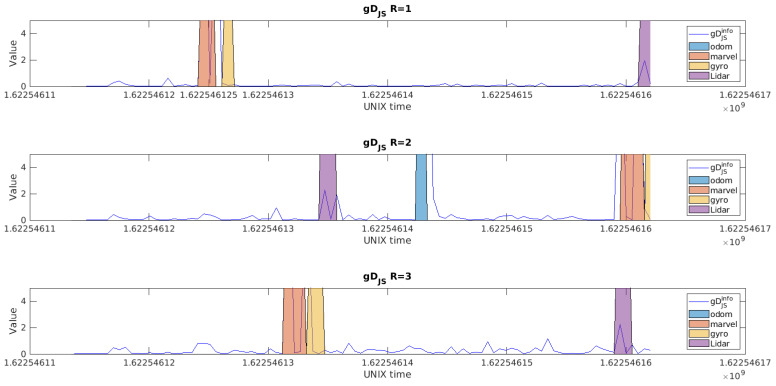
Detection Residuals for $traj_{5}$ scenario = 8.

**Figure 6 sensors-23-07351-f006:**
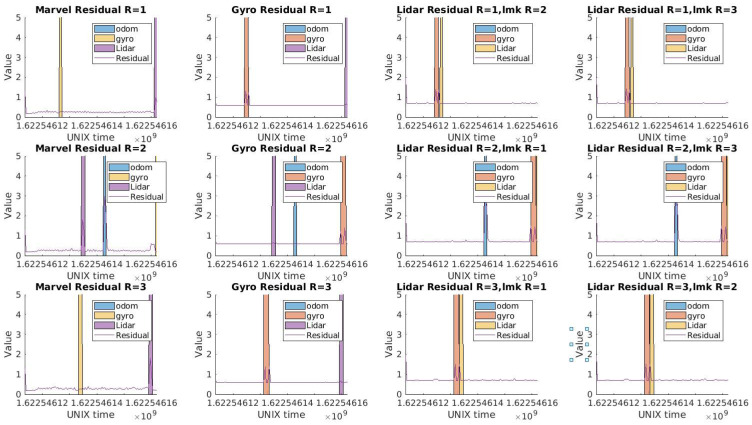
Isolation Residuals for $traj_{5}$ scenario = 8.

**Figure 7 sensors-23-07351-f007:**
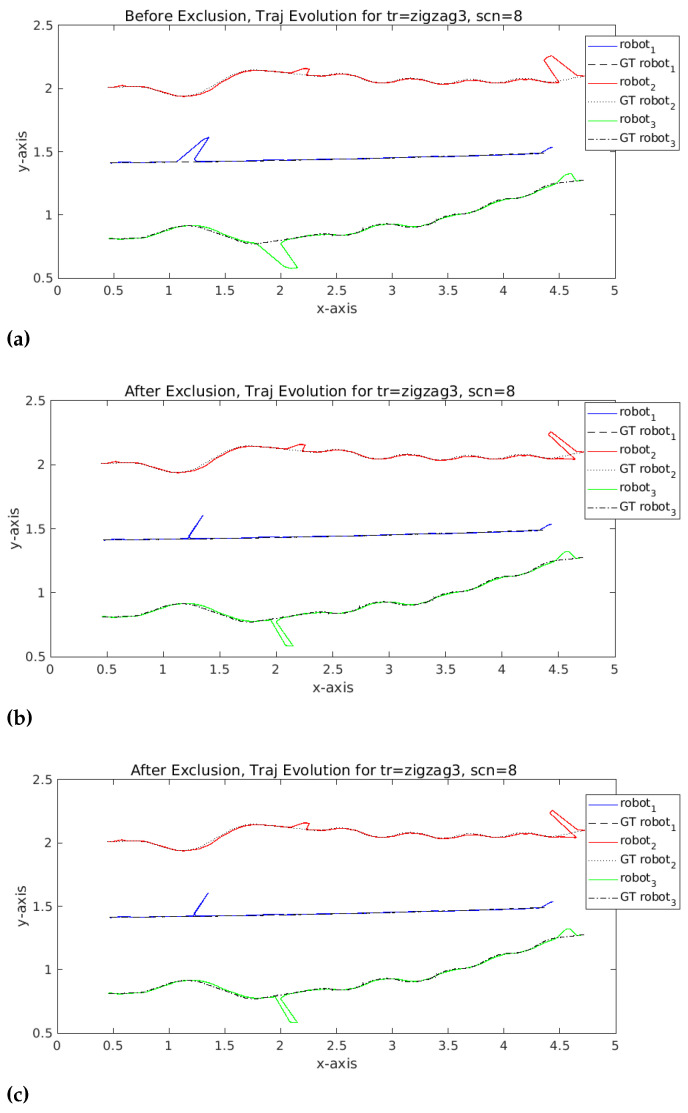
$Trajectory_{5}$ evolution: (**a**) Before exclusion, (**b**) After exclusion centralized, (**c**) After exclusion federated, showing identical performance of exclusion for this trajectory.

**Table 1 sensors-23-07351-t001:** Ratio of Performance of the models for detection and isolation.

	Robot 1		Robot 2		Robot 3	
**Trajectory**	**Accuracy**	**f1-Score**	**Accuracy**	**f1-Score**	**Accuracy**	**f1-Score**
$traj_{1}D-Centr$	0.9250	0.6800	0.9500	0.6536	0.9250	0.6800
$traj_{1}D-Robot$	0.9125	0.6583	0.9500	0.6536	0.9375	0.7056
$traj_{1}D-Fed$	0.9250	0.6800	0.9500	0.6536	0.9250	0.6800
$traj_{1}I-Centr$	0.9625	0.7054	0.9843	0.7876	0.9624	0.6751
$traj_{1}I-Robot$	0.9593	0.6707	0.990	0.81	0.9718	0.7177
$traj_{1}I-Fed$	0.968	0.685	0.99	0.922	0.968	0.694
$traj_{2}D-Centr$	0.9638	0.7762	0.9457	0.6858	0.9457	0.7210
$traj_{2}D-Robot$	0.9698	0.8254	0.9457	0.6858	0.9457	0.7210
$traj_{2}D-Fed$	0.9638	0.7762	0.9457	0.6858	0.9457	0.7210
$traj_{2}I-Centr$	0.97138	0.6829	0.983	0.717	0.9713	0.691
$traj_{2}I-Robot$	0.9728	0.67	0.981	0.628	0.966	0.698
$traj_{2}I-Fed$	0.974	0.658	0.975	0.618	0.9743	0.6913
$traj_{3}D-Centr$	0.9504	0.7092	0.9504	0.7369	0.9257	0.5857
$traj_{3}D-Robot$	0.9455	0.6962	0.9504	0.7369	0.9356	0.6008
$traj_{3}D-Fed$	0.9504	0.7092	0.9504	0.7369	0.9207	0.5791
$traj_{3}I-Centr$	0.9789	0.619	0.974	0.607	0.9678	0.554
$traj_{3}I-Robot$	0.975	0.60	0.974	0.597	0.9777	0.577
$traj_{3}I-Fed$	0.977	0.619	0.97	0.606	0.971	0.576
$traj_{4}D-Centr$	0.9032	0.6278	0.9569	0.8218	0.9462	0.7081
$traj_{4}D-Robot$	0.9032	0.6278	0.9569	0.8218	0.9462	0.7080
$traj_{4}D-Fed$	0.9032	0.6278	0.9569	0.8218	0.9462	0.7080
$traj_{4}I-Centr$	0.954	0.687	0.973	0.76388	0.9677	0.6964
$traj_{4}I-Robot$	0.9596	0.7817	0.9811	0.7826	0.975	0.722
$traj_{4}I-Fed$	0.9596	0.739	0.9704	0.709	0.973	0.69
$traj_{5}D-Centr$	0.9484	0.63	0.9278	0.6626	0.9278	0.5922
$traj_{5}D-Robot$	0.9485	0.6295	0.9278	0.6626	0.9278	0.5922
$traj_{5}D-Fed$	0.9485	0.6294	0.9278	0.6626	0.9278	0.5921
$traj_{5}I-Centr$	0.987	0.788	0.966	0.708	0.971	0.653
$traj_{5}I-Robot$	0.987	0.796	0.966	0.605	0.9742	0.6541
$traj_{5}I-Fed$	0.984	0.7877	0.961	0.676	0.976	0.779

## Data Availability

Data is the property of CRIStAL Laboratory and cannot be shared publicly at this time.

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
