# Peer review of "Comparative Analysis of Centralized and Federated Learning Techniques for Sensor Diagnosis Applied to Cooperative Localization for Multi-Robot Systems"

_sensors, 2023, doi:10.3390/s23177351_

Round 1

Reviewer 1 Report

The article is interesting for goup-robots designer. 

Scientific novelty includes original methods for determining inoperable elements in a robot by taking into account information in a group

The accurate localization of multiple robots in a cooperative setting is crucial for the success of various applications in robotics. However, sensor faults can significantly degrade the performance and reliability of a multi-robot localization system. In this paper, we propose a fault-tolerant multi-robot cooperative localization system that incorporates discriminative machine learning techniques to detect sensor faults. 

Remarks:

1  the study background needs to be updated with more recent publications.

in article have 2 of the publication of 2022 (authors-applicants) and 1 publication of 2020

2 it is necessary to formalize the statement of the problem and determine the boundary conditions

3 it is not clear why the authors limited themselves to only 3 robots and it is necessary to justify why it is possible to expand the method to a larger number of robots in a group and what is the maximum number of robots in a group that is acceptable for the method to work.

4 the paper gives performance estimates, but does not indicate the computing facilities on which the calculations were carried out. It was necessary to indicate the formal computational complexity in typical operations and the memory complexity.

5 Accuracy and F1-scores require you to show the original error values - you should give them. The representativeness of the sample and the confidence intervals should also be assessed.

6 the graphs are not informative, some axes are not signed. The choice of a trajectory in the form of an ellipse is not justified - see figure 1.

7 when giving numerical estimates, you should decide on the number of significant digits

Author Response

Greetings to you,

First, I would like to thank you for the time and effort in reading and directing the work in order to improve it. I pretty much appreciate it, and thank you again for the constructive comments, that I will answer in the following. The comments were taken into consideration in the new submission.

  1. Concerning the references, you are absolutely right, they are rather old, approximately 20 years ago. This was rectified with more recent references in the new updated version.
  2. The assumptions and limitations are mentioned in the second paragraph for the problem formulation, they were reformulated and expanded.
  3. The study is limited to three robots in validation to the cooperation hypothesis of having  greater or equal to 2 robots, and adding one robot above this limit to validate the hypothesis of expansion.
  4. The learning and testing are done offline on a pre-recorded database, the PC used for the data generation, training and testing is 16 GB RAM core-i5 PC under ubuntu 20.04 LTS operating system. This specification was added in the new version of the submission.
  5. Concerning the original error and the use of the metrics :
    1. The chosen severity for this study is 30%, and the duration of fault is between [1%, 3\%] of the total time of the trajectory, making the faults brief and low. The scenarios were generated in a way for the sensor faults to occupy different instances each time, and in different order of occurrence.
    2. The chosen comparison metrics for this study are accuracy and F1-score, to measure how well the model correctly predicts the class labels of the samples in the dataset, and combine in a single metric both precision and recall to provide a balanced measure of a model's performance, especially when dealing with imbalanced datasets or scenarios where false positives and false negatives have different implications.
  6. The trajectories were generated to encompass scenarios in which multiple robots can coexist within an environment and mutually observe one another. The selection of circular or transversal shapes for these trajectories represents the fundamental geometric forms underlying their structure. 
  7. We consider the measures for comparison up to the fourth decimal place (10^(-4)), as this is where most of the significant changes occur and provides a meaningful basis for comparing the resultant numbers.

I repeat my expression of gratitude for your interest and your attentive reading.

Best regards,

Zaynab EL MAWAS

Reviewer 2 Report

This work presented a Comparative analysis of centralized and federated learning techniques for sensor diagnosis applied to cooperative localization for multi-robot systems. Overall the presented work is interesting to the readers. Our comments are given below:

1. In the abstract section, even though I can roughly get the idea of what the authors are trying to convey, however, the abstract is too boring and needs to be revised.
2. The objectives of this work must be carefully revised based on the discussion in the later sections and simulation section.
3. The literature review can be improved further by adding recently published novel studies since the existing specific contributions or findings are not highlighted. Also, you need to investigate works like in 
https://doi.org/10.1016/j.dib.2021.106854 from different points of view.

4.  Algorithm 1 must be explained in the text.

5- The results were poorly discussed in light of the previous research findings. Please put results in the context of what was known before and then discuss potential challenges, future directions/recommendations, and why these kinds of studies are important!

Moderate

Author Response

First, I would like to thank you for the time and effort in reading and directing the work in order to improve it. I pretty much appreciate it, and thank you again for the constructive comments, that I will answer in the following. The comments were taken into consideration in the new submission.

  1. I have rewritten the abstract upon your request, hopefully it is more appealing.
  2. A paragraph was added to clearly state the contributions of this work
  3. Our work concerns the creation of fault tolerant localization solutions, where we use sensors as means to observe the environment and localize the vehicle accordingly. The sensors are prone to errors, and we investigate the possible ways to detect the presence of faults on those sensors in order to exclude them from the data fusion step to increase the precision of the position estimation. The paper that you mentioned does not concern this case of study.
  4. The explanation of algorithm 1 was added in the new submission
  5. The results discuss the comparison metrics, training time and storage that it occupies, as well as the results after the actual implementation of the algorithm and the improvement they provided on the localization solution. More interpretations were added regarding the parameters.

I repeat my expression of gratitude for your interest and your attentive reading.

Best regards,

Zaynab EL MAWAS

Round 2

Reviewer 1 Report

ok